# The Comparison of Fatty Acid Composition and Lipid Quality Indices of Roach, Perch, and Pike of Lake Gusinoe (Western Transbaikalia)

**DOI:** 10.3390/ijerph18179032

**Published:** 2021-08-27

**Authors:** Selmeg V. Bazarsadueva, Larisa D. Radnaeva, Valentina G. Shiretorova, Elena P. Dylenova

**Affiliations:** Baikal Institute of Nature Management of Siberian Branch of the Russian Academy of Sciences, Sakhyanovoy St., 6, 670047 Ulan-Ude, Russia; radld@mail.ru (L.D.R.); vshiretorova@rambler.ru (V.G.S.); edylenova@mail.ru (E.P.D.)

**Keywords:** freshwater fish, Lake Gusinoe, fatty acids, lipid quality indices

## Abstract

This paper describes the study of the fatty acid (FA) composition of three fish species (roach, perch, and pike) from Lake Gusinoe (western Transbaikalia). Using principal component analysis, the fatty acid composition of the studied fish species was shown to be species specific. The muscle tissue of roach, perch, and pike was found to contain high levels of polyunsaturated fatty acids (PUFA), including essential docosahexaenoic (DHA), eicosapentaenoic (EPA), and arachidonic acids. Indicators of nutritional quality based on the fatty acid composition showed that the values of the hypocholesterolemic/hypercholesterolemic (HH) ratio indices were sufficiently high. The atherogenicity (AI) and thrombogenicity (TI) indices, which are indicators for the nutritional value, were less than 1 in the studied fish. In terms of flesh lipid quality (FLQ), pike and perch had the highest proportion of total EPA + DHA. According to the obtained data for the composition of fatty acids in the muscle tissue of the studied fish from Lake Gusinoe, the anthropogenic load exerted on Lake Gusinoe has not yet statistically significantly affected the fish muscle quality.

## 1. Introduction

Lake Gusinoe is the largest freshwater lake among all water bodies in the Transbaikal region and one of the most important in terms of the intensity of water utilization for fish husbandry. It is also a valuable fishery reservoir where current fish gross production is estimated to be around 250–350 tons per year [1]. More than half a century of research on the species composition of fish in Lake Gusinoe has found evidence of 23 fish species from 11 families inhabiting the lake at different periods of its existence. According to recent long-term studies, 15 fish species are currently inhabiting Lake Gusinoe [1]. Mass death of perch and an ulcerative disease of pike were determined to have occurred in 1974 [2,3,4]. An increase in the chemical pollution of Lake Gusinoe was found to have occurred in 1978–1985, coinciding with when the Gusinoozersk State Regional Power Plant (SRPP) and the Kholboldzhinsky opencut coal mine began operating [1,4,5]. The Gusinoozersk SRPP discharges about 2 million m^3^ of heated water into the northern part of the lake every day [6]. More than 600 million m^3^ of water from the lake has been used for technical needs each year, which corresponds to one-quarter of its volume. According to the data of the Buryat Centre of Hydrometeorology and Environmental Monitoring [7], the water quality in this lake balances between the categories “polluted” and “very polluted” at the observation point of Lake Gusinoe Station. There is a constant increase in anthropogenic pressure. For example, discharge of only partially clean wastewater into the lake after the cooling of equipment without purification of the Gusinoozersk SRPP increased from 289 million m^3^ in 2009 to 400 million m^3^ in 2014; the discharge in 2016 was 431 million m^3^. As a result, a deterioration in water quality due to the presence of specific pollutants and deterioration of the oxygen regimen is observed [7]. The initiation of ichthyofauna depauperation has previously been noted, and it influences the degradation of fish populations in the water body, a process that has continued to the present time. In 2000–2006, pike was infrequently detected in commercial landings [2]. It has been found that the growth rate of perch in Lake Gusinoe is 11% higher than the average for water bodies within the Transbaikal region and has reached the levels of previous years. In roach populations, small and medium sizes dominate over large ones, and in general, this indicates the formation of a favorable situation for the roach population in Lake Gusinoe [8].

Among the various available bioindicators of surface water contamination, fish are one of the most suitable objects for assessing the quality of aquatic systems. Water quality assessment and the state of hydrobionts, including fish, which are considered as objects of human nutrition, indicate a need to provide ongoing environmental monitoring. According to the classification of water by the frequency of pollution events and the frequency in which maximum permissible concentrations are exceeded, in Lake Gusinoe, the typical pollution of the low and average levels has been registered with resistance to oxidation organic substances and copper, the stable low level with easily oxidable organic substances and the unstable low level with zinc [7]. The heavy metal content of fish inhabiting Gusinoe Lake was investigated most recently in 2003 [9] and earlier in 1992. This revealed that the heavy metal content of fish in 2003 did not exceed the maximum permissible concentration and had decreased compared to the data of 1992. Polyaromatic hydrocarbon pollution of sediments and surface water of Lake Gusinoe is relatively low. It is comparable to the level of pollution in the Arctic seas. It is mainly caused by local sources of a pyrolytic nature, particularly coal firing by industrial enterprises and wood burning for heating [10].

The toxic effects can be inferred at organ, tissue, cellular, subcellular, genetic, and other levels. As a result, protein, lipid, and fat metabolism are significantly disrupted [11]. The most important physiological and biochemical indicators of the state of organisms and populations under different habitat conditions are lipid indicators. Lipids are the main source of energy for organisms, and the functioning of any ecosystem is largely associated with their biosynthesis and transport in food chains [12,13,14]. The main type of fat playing a role in the dietary patterns of contemporary people is n-6 PUFA from terrestrial mammals, but the consumption of n-3 PUFAs is inadequate, even in highly developed countries [15,16,17]. Consumption of fish is known to provide many benefits for human health due to its high content of essential PUFA of the n-3 family, namely EPA and DHA (22:6n3). Regular consumption of essential fatty acids prevents cardiovascular diseases and neural disorders [18,19,20,21]. Thus, this determines the importance of biochemical studies of fish in the Baikal region, not only in terms of studying ecology and taxonomy but also for clarifying their nutritional value.

Thus, the aim of the study was to determine the fatty acid composition and quality indicators of the muscle tissue of roach *(Rutilus rutilus* Linnaeus), perch (*Perca fluviatilis* Linnaeus), and pike (*Esox Lucius* Linnaeus) of Lake Gusinoe (western Transbaikalia).

## 2. Materials and Methods

### 2.1. Field Sampling

The sampling of roach (*n* = 12), perch (*n* = 12), and pike (*n* = 10) from the water of Lake Gusinoe (Figure 1) was conducted in July 2018 using a trap net at a depth of about 2–8 m. Mature adult fish were analyzed, with a length (including the caudal fin) ranging from 21 to 24 cm for roach, 22 to 29 cm for perch, and 48 to 53 cm for pike. The collected fish were stored at −18 °C for less than seven days prior to laboratory analyses.

### 2.2. Collection Site

Lake Gusinoe is the second largest lake in the Baikal Natural Territory after Lake Baikal. It is located in the Gusinoozersk intermountain basin, which has a tectonic origin. The catchment area of Lake Gusinoe is 924 km^2^. The lake surface covers 164 km^2^. It has a maximum depth of 26 m (average 15 m), length of 25 km, and average width of 8 km. The long-term average water mass volume is 2.4 km^3^. The maximum amplitude of level fluctuations reaches 95 cm. There are both seasonal (maximum in July–August) and secular (3–5 m) variations. The lake is stretched from southwest to northeast and has an oval-kidney shape. The structure is a trough-type pan with two unequal basins [4].

Lake Gusinoe belongs to a low-flow reservoir with a slow water exchange system. The inflows are shallow, and they tend to freeze in winter, but in summer, they often do not reach the lake, flowing out at a distance of 300–500 m from the shore in loose sediments. These flows occur as small mountain rivers and streams flowing from the Khambinsky ridge. The majority of the lake water is brought by the Tsagan-Gol River, which is a sleeve of the Temnik River. The Zagustay River is a second inflow in terms of water content, and it flows into the northern part of the lake. The inflows are fed by rain. The Bayan-Gol River flows from the southeastern part of Lake Gusinoe. It flows into the Selenga River after passing through several small lakes located in the Tamchinskaya steppe [4]. The Bayan-Gol River has a constant flow throughout the year and does not freeze in winter.

Lake Gusinoe is the main source of drinking and household water supply for more than 31,000 residents of Gusinoozersk city, Gusinoe Lake and Baraty villages, Murtoy station, and a number of offshore bases and stations. At the same time, the lake serves as a receiver of purified effluent from the biological treatment plant of Gusinoozersk city and wastewater without biological treatment of Gusinoe Lake village. Mechanical and biological treatments are used in waste treatment facilities, after which the water is chlorinated and discharged into the lake. Low wastewater quality is caused by the overloading of waste treatment facilities, especially during winter. Mine and quarry waters of the Kholboldzhinsky opencut coal mine have been discharged into Lake Gusinoe for more than 40 years. In recent decades, the lake has been used as a cooling reservoir for the Gusinoozersk SRPP. Discharge from slag disposal and rainfall from the atmospheric emissions of the thermal power plant industrial site have a significant impact on the hydrochemical and hydrobiological regime of the water body and its sanitary state [1]. 

### 2.3. Sample Derivatization

The FA contents were determined in subsamples of the tissues weighing approximately 0.5–1.0 g. The subsample was cut from the white dorsal muscle, from the side about 2/5 of the distance from snout to tail, carefully avoiding red muscle, skin, and bone. The weighed and homogenized subsamples of fish muscle tissues were transferred to 15 mL thick-walled glass tubes. After the addition of 1 mL of anhydrous methanol containing 2 M HCl and exchange of the atmosphere in the tubes with nitrogen gas, the tubes were securely closed with Teflon-lined screw caps and placed in an oven for 2 h at 90 °C for complete methanolysis [22,23]. After cooling to room temperature, the tubes were opened and the methanol evaporated down to about 0.5 mL under a stream of nitrogen gas, and 0.5 mL of distilled water was then added to reduce the solubility of the formed FA methyl esters (FAME), which were extracted with 2 × 1 mL *n*-hexane. All chemicals were of high purity.

### 2.4. Fatty Acid Analysis

FAs were analyzed by gas chromatography–mass spectrometry (GC-MS) as described in [24]. Methyl esters of FAs were examined by the gas chromatography–mass-spectrometry method using an Agilent 6890B gas chromatograph with a 5973 N quadrupole mass spectrometer as the detector and an HP-5MS capillary column (30 m × 0.25 mm × 0.2 µm; Hewlett-Packard, Palo Alto, CA, USA). Helium (99.9999% purity) was used as the carrier gas at a flow rate of 1.5 mL/min. The oven temperature was programmed as follows: it was kept at a constant temperature of 125 °C for 0.5 min, from 125 °C to 320 °C (at the rate of 7 °C/min), and kept constant at 320 °C for 0.5 min. The injector and detector temperatures were set to 280 °C and 250 °C, respectively. The split ratio was adjusted to 40:1. For quantitative analysis, the characteristic ions and retention times of all analytes were determined in full-scan (*m*/*z* 50 to 550) using electron ionization (electron energy—70 eV). The MS data were acquired in scanning mode at a speed of 2.5 s per scan. The percent composition of the lipid fraction derivatives was calculated from the GC peak areas relative to the total peak area based on the GC-MS analyses of the lipid fraction. Qualitative analysis was based on comparing the retention times and total mass spectra of the corresponding pure compounds using NIST14.L and standard mixtures of Bacterial Acid Methyl Esters (CP Mix, Supelco, Bellefonte, PA, USA) and Fatty Acid Methyl Esters (Supelco, 37 compounds, FAME Mix, 10 mg/mL in CH_2_Cl_2_). The relative amount of each FA in a sample was expressed as a percent of the sum of all fatty acids in the sample.

### 2.5. The Lipid Quality Indices

The fatty acid composition was used to determine several nutritional parameters of lipids in fish muscles. The lipid quality indices were calculated using the following equations. 

#### 2.5.1. Index of Atherogenicity

The index of atherogenicity (AI) is the correlation between the total main saturated fatty acids and the main classes of unsaturated fatty acids. The former is considered to be proatherogenic, as it provides the adhesion of lipids to cells of the immunological and circulatory system. The latter are antiatherogenic, as they inhibit the aggregation of plaques and diminish the levels of esterified fatty acid, cholesterol, and phospholipids, thereby preventing the appearance of micro and macro coronary diseases [25,26,27,28,29].
(1)$$AI=\frac{C12:0+\left(4\times C14:0\right)+C16:0}{\left(n-3\right)PUFA+\left(n-6\right)PUFA+MUFA}$$
where PUFA—polyunsaturated fatty acids, MUFA—monounsaturated fatty acids, C12:0—lauric acid, C14:0—myristic acid, and C16:0—palmitic acid.

#### 2.5.2. Index of Thrombogenicity

The thrombogenicity index (TI) is defined as the relationship between the pro-thrombogenetic (saturated) and the anti-thrombogenetic fatty acids (MUFA, n-6 PUFA, and n-3PUFA) [25,28].
TI=C14:0+C16:0+C18:00.5×C18:1+0.5×sum of other MUFA+0.5×(n−6)PUFA+3×(n−3)PUFA+(n−3)PUFA(n−6)PUFA

#### 2.5.3. Flesh Lipid Quality

The flesh lipid quality (FLQ) is the percentage correlation between the main n-3 PUFA (EPA + DHA) and the total lipids. Thus, the higher the value of this index, the better the quality of the dietary lipid source [28,29,30]: (2)$$FLQ=100\times \frac{EPA+DHA}{\%\;of\;total\;fatty\;acids}$$
where hypocholesterolemic fatty acids (OFA): OFA = C12:0 + C14:0 + C16:0, hypercholesterolemic fatty acids (DFA): DFA = C18:0 + UFA, EPA—eicosapentaenoic acid (C20:5), DHA—docosahexaenoic (C22:6), UFA—unsaturated fatty acids (MUFA + PUFA), and C18:0—stearic acid

### 2.6. Statistical Analysis

The results are presented as arithmetic mean ± standard deviation (SD). The calculation of mean, SD, and Fisher’s LSD test was carried out by the Microsoft Excel software. Differences were found to be significant at *p* ≤ 0.05.

The FA composition data were subjected to multivariate statistical analysis using principal component analysis (PCA) to obtain a clear picture of the differences in the FA composition between fish species and their tissues. All statistical analyses were conducted using Sirius ver. 8.5 software [31,32]. FAs found in all or the majority of samples were subjected to statistical analysis, and their relative values were logarithmically transformed to avoid domination of the most abundant FAs. PCAs were performed by positioning the samples in a 24-dimensional space described by the variables (FAs). New coordinate PCs were drawn through the centroid of the samples in such a way that principal component 1 (PC1) expressed the direction of the largest, and principal component 2 (PC2, which is orthogonal to PC1) was the direction of the second largest variation among all samples. The original variables (FAs) were displayed together with the samples, resulting in a biplot visualizing the correlation between samples and FAs. FAs that were far from the origin contributed most to sample variation. In addition, samples that were close to an FA along a principal component contained relatively more of this FA than samples lying in the opposite direction. 

## 3. Results

### 3.1. Fatty Acids

A total of 37 fatty acids were found in the muscle tissue of the studied fish species. Acids in relative proportions higher than 0.1% are presented in Table 1. Palmitic 16:0 and stearic 18:0 acids were dominant among the saturated fatty acids (SFA). Such saturated fatty acids with odd-numbered carbon atoms, such as 15:0, iso15:0, iso17:0, and aiso17:0, were found in fewer amounts. Oleic acid (18:1n9) was the dominant MUFA in all species. The content of oleic acid was found to be negatively correlated with the content of linoleic and α-linolenic acids. Docosahexaenoic (22:6n3), arachidonic (20:4n6), and eicosapentaenoic (20:5n3) acids were dominant among the PUFAs. 

Multivariate analysis of the main components allowed us to establish the differences revealed in the fatty acid composition of fish samples from Lake Gusinoe (Figure 2). 

### 3.2. Lipid Quality Indices

There were no significant differences between the values for the index of atherogenicity (AI), index of thrombogenicity, and hypercholesterolemic (OFA) and hypocholesterolemic acids (DFA) when comparing the muscles of the studied fish (Table 2). The muscles of pikes showed a significantly higher flesh lipid quality index (FLQ) (41.25) (*p* ≤ 0.05) than those of roach (34.75) and perch (30.11).

## 4. Discussion

The characterization of tissue metabolism processes can serve as the fastest and most effective means of assessing toxicant effects. The intensification of free radical oxidation of lipids and the dysfunction of antioxidant systems are important responses to the toxic effect of pollutants that occur in living organisms. Many researchers studying the processes of lipid peroxidation affected by xenobiotics, as well as the thermal pollution of water bodies, found that the main reaction of aquatic organisms is the enhancement of free-radical lipid oxidation [33,34]. We investigated three fish species of different trophic levels, which are the most popular among locals and fishermen: roach *Rutilus rutilus* L. (a plankton- and benthos-feeding fish), which has ontogenesis development mostly in water surface layers, and predators perch *Perca fluviatilis* L. (with a diet of benthic organisms along with fish) and pike *Esox lucius* L. (exclusively fish eating). Nowadays, there is a significant anthropogenic load on Lake Gusinoe, but the fatty acid composition and lipid quality indices of fish inhabiting Lake Gusinoe have not previously been studied.

Among the fatty acids, highly unsaturated n-3 fatty acids or long-chain n-3 PUFA, particularly EPA and DHA, influence human health, early development, and the prevention of some diseases; therefore, dieticians recommend consuming foods containing these fatty acids. The n-3 PUFA contents in the muscle tissue of pike (45.2%) were higher (*p* ≤ 0.05) than in the muscle tissue of roach and perch (38.1% and 39.3%, respectively). However, the muscles of roach and perch had significantly higher contents of n-6 PUFA (13.7% and 12.1%, respectively (*p* ≤ 0.05)). There were no significant differences between the values of the sum of SFA and UFA in the muscles of the examined fish (*p* > 0.05). The sum of MUFA in pike was significantly lower than for the other two fish species (*p* ≤ 0.05). The fatty acid profiles of the fish studied in this research are similar among the same fish species [35,36,37], which probably indicates that the fish species of Lake Gusinoe are not significantly influenced by anthropogenic impact. 

Long-chain PUFAs of n-3 (in particular EPA and DHA) and n-6 (arachidonic acid) families, among all fatty acids, are considered to be physiologically valuable for fish [38,39]. According to our research, these essential acids dominated among PUFAs. The content of arachidonic 20:4n6 and eicosapentaenoic 20:5n3 acids were higher in roach than in perch and pike (*p* ≤ 0.05). 

It was revealed that among other PUFAs, the content of linolenic and linoleic acids was relatively higher in the planktobenthos-eating species roach and in pike (*p* ≤ 0.05), which is characterized by habitation at the bottom of the water body in thickets of aquatic vegetation. Its nutrition is also associated with benthic hydrobionts. These acids were dominant among the PUFAs in macrophytes of Lake Gusinoe [40]. This lake belongs to oligotrophic lakes [41,42], where the taxon of cryptophytic, dinophytic, and diatom algae are inhabitants, synthesizing EPA and DHA in large quantities [43,44]. It is accepted that fish inhabiting such water bodies may have a higher nutritional value in terms of PUFA content [44].

Polyunsaturated n-3 fatty acids are essential for fish. This has been established as a result of numerous experiments, including those on freshwater species [45]. An increased content of n-6 and n-3 PUFA identified in this research in the muscles of the studied fish determines their high nutritional value. The ratio of fatty acids n-3/n-6 in the muscle tissue of the studied fish was 2.8–4.6. The n-3/n-6 ratios are typical for freshwater fish, where the ratio lies in the interval 0.5–3.8, compared to the interval 4.7–14.4 for marine fish [46]. From a nutritional point of view, the ratio is close to the recommended ideal value of between 0.5 and 1 [47]. Therefore, freshwater fish are currently recognized as a valuable dietary component of human nutrition, as valuable as sea fish [48]. 

According to the load graph (Figure 2), a group of perch samples had higher amounts of 16:1n9 and 20:1n9 acids and the lowest amount of linolenic 18:3n-3 acid. The content of oleic 18:1n9 acid in the muscles of perch was much lower than that in roach (*p* ≤ 0.05) (Table 1). Significantly lower amounts of 16:1n7 and 20:4n6 were observed in pike muscle tissue in comparison with other fish species (*p* ≤ 0.05). Roach samples were characterized by the lowest content of DHA 22:6n-3. The muscles of the roach had the highest percentage of C15–17 branched FAs. We found that the fish-eating species, perch and pike, had lower levels of 18:2n-6 and 20:5n-3 acids but significantly higher levels of 22:6n-3 (*p* ≤ 0.05) in tissues compared to the planktobenthos-eating species roach. The levels of other acids among the studied fish species did not differ crucially (*p* > 0.05). Thus, the use of multivariate analysis allowed us to determine the species specificity of the fatty acid composition.

The indicators of nutritional quality based on the FA composition were determined. Two indicators, AI and TI, were investigated, as their effect on the frequency of pathogenic events, such as atheroma and/or thrombus formation, differs from the formation of single FAs. The atherogenic and thrombogenicity indices were not significantly different (*p* > 0.05) between the different fish, and their values were 0.36–0.38 and 0.18–0.21, respectively. The results of the current research on AI and TI are consistent with studies conducted by Linhartová et al. [49], which found that the indicators in all analyzed fish, except for *Nile tilapia*, were below 0.5, which indicates significant benefit to human health if these fish are included in the human diet. The consumption of foods or products with a lower AI can reduce the levels of total cholesterol and LDL-C in human blood plasma [50]. The TI characterizes the thrombogenic potential of FAs, indicating the tendency to form clots in blood vessels and provides the contribution of different FAs, which denotes the relationship between the pro-thrombogenic FAs (lauric C12:0, myristic C14:0 and palmitic C16:0 acids) and the antithrombogenic FAs (MUFAs and the n-3 and n-6 families). Therefore, the consumption of foods or products with a lower TI is beneficial for cardiovascular health. 

The HH index provides insight into the effect of FA on blood cholesterol level. A higher value of the HH index is preferable. The contents of hypocholesterolemic and hypercholesterolemic fatty acids in the muscles of the studied fish were not different as follows: pike (21.74% and 71.50%), perch (21.56% and 70.68%), and roach (21.31% and 71.81%) (*p* > 0.05). According to our research, the HH values ranged from 3.28 to 3.29 for perch and pike and to 3.37 for roach. The HH ratio may more accurately reflect the effect of the FA composition on cardiovascular health. For fish, the value ranges from 1.54 to 4.83, with the exception of *Opisthonema oglinum*, which has an HH value of 0.87 [51].

The FLQ is a more suitable index assessment of marine products given their higher proportions of EPA and DHA. The FLQ index was 41.25% in pike muscle tissue, 37.72% in perch, and 34.75% in roach. Tonial et al. (2014) suggested that MUFA and PUFA are more beneficial for health as they prevent coronary heart disease [52].

## 5. Conclusions

The fatty acid composition of the studied fish species is species specific, which was shown by principal component analysis. The fatty acid composition of the muscle tissue of roach, perch, and pike contains high levels of PUFA, including essential docosahexaenoic, eicosapentaenoic, and arachidonic acids. It was revealed that the ratio of n-3/n-6 PUFAs in the studied freshwater fish was 2.8–4.6, which is typical for freshwater fish. Indicators of nutritional quality based on the FA composition showed that the values of the HH indices were sufficiently high, and the AI and TI indices, which indicate the nutritional value of the studied fish, were less than 1. In terms of FLQ, pike and perch had the greatest amount of total ratio EPA + DHA. According to the data obtained for the composition of fatty acids in the muscle tissue of the studied fish from Lake Gusinoe, the anthropogenic load exerted on Lake Gusinoe has not yet statistically significantly affected the state of the fish in terms of their nutritional quality. The monitoring of Lake Gusinoe, in particular, the study of the content in fish of such pollutants as heavy metals, phthalates, and pesticides, as well as their effect on the fatty acid composition, will be carried out by us in the near future.

## Figures and Tables

**Figure 1 ijerph-18-09032-f001:**
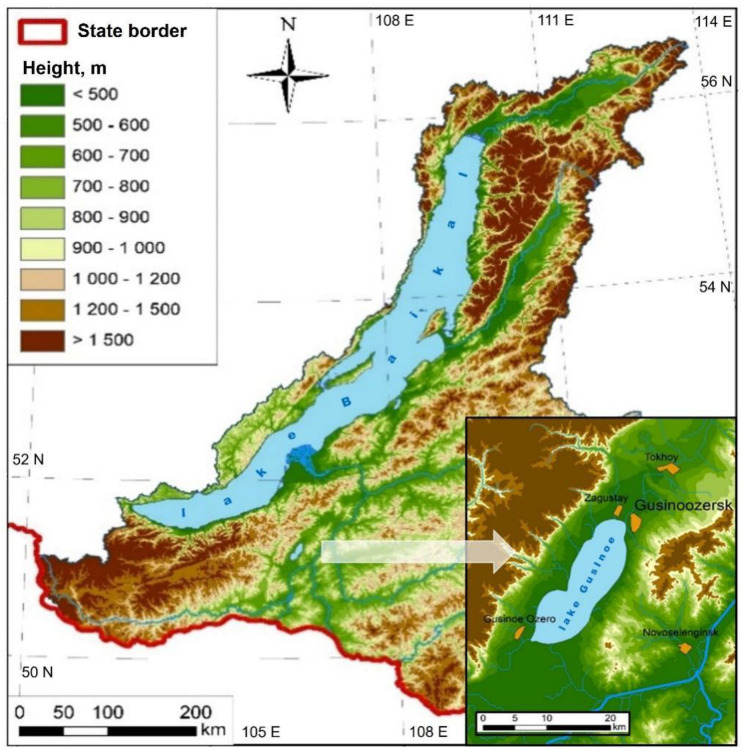
The studied area was located in western Transbaikalia, Russia (geographical coordinates: 51°06′51″ N, 106°15′41″ E).

**Figure 2 ijerph-18-09032-f002:**
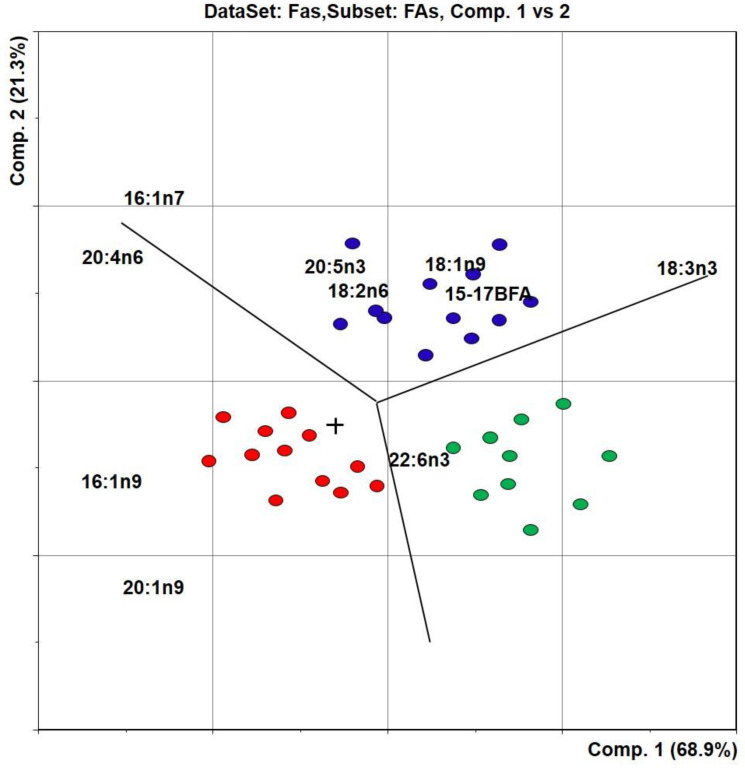
Data distribution of the studied samples according to the content of main fatty acids in the muscle tissue of the studied fish species: blue—roach, red—perch, and green—pike.

**Table 1 ijerph-18-09032-t001:** Fatty acid composition (% of total fatty acids) of the muscles of the studied fish (mean ± SD).

Fatty Acids *	Roach (*n* = 12)	Perch (*n* = 12)	Pike (*n* = 10)
12:0	0.15 ± 0.02 a	0.14 ± 0.01 a	0.14 ± 0.02 a
14:0	0.82 ± 0.16 a	0.89 ± 0.14 ab	1.04 ± 0.17 ab
15:0	0.34 ± 0.05 b	0.34 ± 0.05 ab	0.42 ± 0.07 a
16:0	20.34 ± 3.10 a	20.53 ± 2.44 a	20.56 ± 2.67 a
15–17 BFA **	1.22 ± 0.24 a	0.65 ± 0.10 ab	0.44 ± 0.06 b
17:0	0.54 ± 0.08 a	0.38 ± 0.06 a	0.32 ± 0.05 a
18:0	6.20 ± 0.83 a	5.04 ± 0.66 a	6.09 ± 0.74 a
16:1n9	1.44 ± 0.14 b	2.67 ± 0.34 a	1.05 ± 0.14 b
16:1n7	3.34 ± 0.46 a	3.49 ± 0.52 a	1.53 ± 0.24 b
18:1n9	8.67 ± 0.69 a	6.88 ± 0.82 b	7.68 ± 0.90 ab
20:1n9	0.34 ± 0.06 b	0.64 ± 0.10 a	0.13 ± 0.01 c
18:3n3	1.82 ± 0.3 a	0.91 ± 0.11 b	2.14 ± 0.33 a
18:4n3	0.23 ± 0.02 a	0.13 ± 0.01 a	0.34 ± 0.03 a
18:2n6	4.08 ± 0.64 a	2.74 ± 0.51 b	2.41 ± 0.40 b
20:2n6	0.29 ± 0.05 ab	0.42 ± 0.07 a	0.12 ± 0.01 b
20:4n6	8.10 ± 0.54 a	7.71 ± 0.44 ab	6.22 ± 0.55 b
20:5n3	12.71 ± 1.68 a	8.07 ± 1.26 b	8.74 ± 1.14 b
20:3n6	0.44 ± 0.07 a	0.24 ± 0.03 b	0.14 ± 0.01 c
20:4n3	0.33 ± 0.05 ab	0.29 ± 0.03 b	0.53 ± 0.08 a
22:5n3	1.05 ± 0.14 a	0.89 ± 0.11 a	0.93 ± 0.12 a
22:5n6	0.78 ± 0.12 a	0.91 ± 0.10 a	0.94 ± 0.11 a
22:6n3	22.04 ± 2.15 b	29.65 ± 3.08 a	32.51 ± 3.14 a
Σ SFA	29.61 ± 2.08 a	27.97 ± 3.10 a	29.01 ± 1.98 a
Σ MUFA	13.79 ± 1.78 a	13.68 ± 1.64 a	10.39 ± 1.25 b
Σ n-3 PUFA	38.13 ± 3.16 b	39.94 ± 3.84 b	45.19 ± 3.95 a
Σ n-6 PUFA	13.69 ± 1.81 a	12.02 ± 1.54 a	9.83 ± 1.06 b
Σ PUFA	51.82 ± 6.72 a	51.96 ± 5.21 a	55.02 ± 7.25 a
Σ UFA	65.61 ± 6.13 a	65.64 ± 6.02 a	65.41 ± 5.89 a
n-3/n-6	2.79 ± 0.24 b	3.32 ± 0.32 b	4.60 ± 0.72 a

* the first figure indicates the number of carbon atoms, the second indicates the number of unsaturated bonds, and the third indicates the first carbon atom with a double bond as counted from the methyl group; ** the total content of branched fatty acids with 15 and 17 carbon atoms in the chain; a–c—significant differences between the fish of the different species (*p* ≤ 0.05). The same letter (in rows) indicates the absence of significant differences (*p* > 0.05).

**Table 2 ijerph-18-09032-t002:** Values for various lipid quality indices for the muscles of the studied fish (mean ± SD).

Indices	Roach (*n* = 12)	Perch (*n* = 12)	Pike (*n* = 10)
AI	0.36 ± 0.02 a	0.37 ± 0.01 a	0.38 ± 0.02 a
TI	0.21 ± 0.01 a	0.19 ± 0.01 a	0.18 ± 0.01 a
FLQ	34.75 ± 1.51 c	37.72 ± 1.36 b	41.25 ± 1.68 a
OFA	21.31 ± 0.82 a	21.56 ± 0.69 a	21.74 ± 1.01 a
DFA	71.81 ± 2.25 a	70.68 ± 1.99 a	71.50 ± 2.56 a
HH	3.37 ± 0.11 a	3.28 ± 0.09 b	3.29 ± 0.08 b

a–c—significant differences between the fish of the different species (*p* ≤ 0.05). The same letter (in rows) indicates the absence of significant differences (*p* > 0.05).

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
