# Peer review of "The Comparison of Fatty Acid Composition and Lipid Quality Indices of Roach, Perch, and Pike of Lake Gusinoe (Western Transbaikalia)"

_ijerph, 2021, doi:10.3390/ijerph18179032_

Round 1

Reviewer 1 Report

I believe my comments are all addressed. For field collection conducted in 2018, I believe this manuscript is good enough for publication.

Author Response

We thank the Reviewer for the valuable advice, suggestions and constructive comments on our manuscript. The comments have been very thorough and useful in improving the manuscript. Undoubtedly, the comments and suggestions have increased the scientific value of revised manuscript.

Reviewer 2 Report

see attached file

Author Response

We thank the Reviewer for the valuable advice, suggestions and constructive comments on our manuscript. The comments have been very thorough and useful in improving the manuscript. About the detail responses, please see the attachment.

Round 2

Reviewer 2 Report

Reviewer comments ijerph-1313407

 “The comparison of fatty acid composition and lipid quality indices of Roach, Perch and Pike of Lake Gusinoe (Western Transbaikalia)”

SV Bazarsadueva, LD Radnaeva, VG Shiretorova and EP Dylenova

The manuscript is the 2. revision of the submission (former ijerph-1238062) describing the fatty acid composition of three fish species (roach, perch and pike) in lake Gusinoe (western Tansbaikalia).

As mentioned in former comments, the total number of fishes per species remains quite low, however it seems this is the first data to build up bigger data basis. Now the manuscript was much improved based on changes in the details on GC-MS investigations (M+M part) and the discussion was condensed and better structured. In addition, the significances were included in tables 1 and 2. In addition, the authors conclude that the monitoring will be continued with a higher number of fishes per species and pollutants will be included in further investigations.

Recommendation:      accept

This manuscript is a resubmission of an earlier submission. The following is a list of the peer review reports and author responses from that submission.

Round 1

Reviewer 1 Report

The manuscript is improved, and, understandably, some fieldwork is not practical to repeat, especially during the pandemic. I have a few additional comments as below.

Major points:

  1. L81, “−25 ° C”: Is this a common practice? I have seen a few references in “-20 ° C”.
  2. L111, “2 M HCl”: What grade is the acid? Trace Metal Grade or ACS? Same comment for all the chemicals?
  3. L267, “18: 3n-3 acid”: It seems 18: 3n-3 does not appear in Figure 2 (capital F for figure on Line 266). Is that because it  is too low to show

Minor points:

  1. L35, “GSRPP”: Note that the abbreviation in Line 34 is “SRPP”. I suggest the authors keep consistency.
  2. L80, “53 sm”: I suppose it should be “cm”?
  3. Table 2: Since the statistical analysis was conducted, it would be better to show a standard error or standard deviation, or 95% confidence interval and specify that the value in the table is mean?

Reviewer 2 Report

The paper describes the fatty acid composition of three fish species (roach, perch and pike) in lake Gusinoe (western Tansbaikalia). According to the reviewer, the manuscript shows some weaknesses. Firstly, the total number of fishes per specie is quite low to get confident statistical statements. However, is seems that no data regarding fatty acids in fish muscle exits until now to get a first data basis?? Based on the ‘polluted history’ of the lake, the results should more discussed in relation to actual general water quality with special focus on pollutant content in water, sediment or at best in fish muscle. Actual, the discussion is poor and too much based on generally knowledge of ‘PUFA effects for human nutrition’. Are there fresh water supply by rivers or other kinds of anthropogenic input?? Secondly, the method description should be extended (sample preparation, GC-MS .. see specific comments). Thirdly, for the assessment of PUFA contribution for human nutrition – the concentration of the fatty acids in fish muscle would be more informative instead of percentage distribution, only.  Last, the English language should be improved (editing service is necessary).

From the reviewer point of view, the topic fits with the scope of the Intern. Journal of Environmental Research und Public Health; however, there are number of comments (see below) that should be addressed before further evaluation of the manuscript.

Recommendation:      major revision

Page    line                  comment

1          19                    …had the highest amount …

1          21                    Better use: ..affected the fish muscle lipid quality

1          34                    …(GSRPP)…

2          77-81               The number of fishes per specie is quite low, and only based on one time

of the year. That means the conclusions are limited, this should be

addressed in the beginning of discussion part. This means the fishes were ‘wild fish’ and not fish from aqua culture – and no information with respect to the diets are available??

3          87                    Are there inflows and outflows to lake Gusinoe – water exchange??

4          111                  The sample procedure from fish to fish muscle tissue including

homogenization steps are missing and should be included.

4          113                  The direct methylation with 2h at 90°C seems questionable with respect to                                the LC-PUFA decomposition , in ref 21 only around 1h was used??? A more

milder lipid extraction and transmethylation procedure is advisable.

4          116                  …with n-hexane.

4          120-138           Why a typical (more polar) GC column (CPSil 88CB) for fatty acid analysis

was not used for better fatty acid separation??

The procedure for fatty acid analysis is poorly described – missing the used

 mode (Scan or SIM –mode), calibration procedure. And, why the results are given in proportion of fatty acids only and not in concentration of fatty acids per 100g fresh fish muscle??

5/6      Table 1                        The significances between the different species were not included!

7          Table 2                        There were no significances included in the table!!

8          228-229           Are there information about the anthropogenic load/impact in water,

sediment or at the best in fish muscle?? Publications regarding pesticides and other pollutants??

8          235-249           This is too general of ‘PUFA knowledge’ for the discussion of your results in

comparison with same fish species in other water bodies.

8          265-266           This is confusing, n3/n6 PUFA ratio low  - higher amount s of n-6 PUFA –

optimal for humans?? Should be clarified!!